Preparation and characterization of vitamin E/calcium/soy protein isolate nanoparticles for soybean milk beverage fortification

Taha Heba A. I. M. 1
Agamy Neveen F. M. 1
Soliman Tarek N. 2
Younes Nashwa M. 3
El-Enshasy Hesham Ali 4 5 6 henshasy@ibd.utm.my
http://orcid.org/0000-0003-3586-1575 Darwish Amira M. G. 7 8 amiragdarwish@yahoo.com
1 Department of Nutrition, High Institute of Public Health, Alexandria University , Alexandria , Egypt
2 Department of Dairy Sciences, Food Industries and Nutrition Research Institute, National Research Centre , Cairo , Egypt
3 Department of Home Economics, Faculty of Specific Education, Alexandria University , Alexandria , Egypt
4 Institute of Bioproduct Development (IBD), Universiti Teknologi Malaysia , Johor , Malaysia
5 Universiti Teknologi Malaysia , Johor , Malaysia
6 City of Scientific Research and Technology Applications , Alexandria, Borg El Arab , Egypt
7 Faculty of Industrial and Energy Technology, Borg Al Arab Technological University BATU , Alexandria, Borg El Arab , Egypt
8 Food Technology Department, Arid Lands Cultivation Research Institute, City of Scientific Research and Technological Applications (SRTA-City) , Alexandria, Borg El Arab , Egypt
Okpala Charles
Electronic publication date: 2024 Apr 3
Publication date: 2024
Volume: 12
Electronic Location ID: e17007
Received 2023 Oct 12; Accepted 2024 Feb 5
Copyright: © 2024 Taha et al.
Copyright year: 2024
Copyright holder: Taha et al.
License: This is an open access article distributed under the terms of the Creative Commons Attribution License, which permits unrestricted use, distribution, reproduction and adaptation in any medium and for any purpose provided that it is properly attributed. For attribution, the original author(s), title, publication source (PeerJ) and either DOI or URL of the article must be cited.
License URL: https://creativecommons.org/licenses/by/4.0/

Keywords: Soybean milk and isolate (SPI), Amino acids, Fatty acids, Phenolic content, Vitamin E/Ca nanoparticles, Vero cells

Funding: UTM through industrial projects R.J130000.7609.4C742 and R.J130000.7609.4C646 This work was financially supported by UTM through industrial projects No. R.J130000.7609.4C742 and R.J130000.7609.4C646. The funders had no role in study design, data collection and analysis, decision to publish, or preparation of the manuscript.

==============================
Soybean milk is a rich plant-based source of protein, and phenolic compounds. This study compared the nutritional value of soybean milk, flour, soy protein isolate (SPI) and evaluated the impact of prepared vitamin E/calcium salt/soy protein isolate nanoparticles (ECSPI-NPs) on fortification of developed soybean milk formulations. Results indicated that soybean flour protein content was 40.50 g/100 g, that fulfills 81% of the daily requirement (DV%), the unsaturated fatty acids (USFs), oleic and linoleic content was 21.98 and 56.7%, respectively, of total fatty acids content. In soybean milk, essential amino acids, threonine, leucine, lysine achieved 92.70, 90.81, 77.42% of amino acid scores (AAS) requirement values respectively. Ferulic acid was the main phenolic compound in soybean flour, milk and SPI (508.74, 13.28, 491.78 µg/g). Due to the moisture content of soybean milk (88.50%) against (7.10%) in soybean flour, the latest showed higher nutrients concentrations. The prepared calcium (20 mM/10 g SPI) and vitamin E (100 mg/g SPI) nanoparticles (ECSPI-NPs) exhibited that they were effectively synthesized under transmission electron microscope (TEM), stability in the zeta sizer analysis and safety up to IC50 value (202 ug/mL) on vero cell line. ECSPI-NPs fortification (NECM) enhanced significantly phenolic content (149.49 mg/mL), taste (6.10), texture (6.70) and consumer overall acceptance (6.54). Obtained results encourage the application of the prepared ECSPI-NPs for further functional foods applications.

Introduction

Plant-based protein beverages have popularity due to their suitability for diets driven by ethical, environmental, or health considerations, such as lactose intolerance (Baroni et al., 2023). Recent market analysis (Zhang, Cai & Ji, 2023) showed that soybean beverages exhibited the highest protein content among plant-based beverages. Furthermore, the addition rate of minerals and vitamins in the examined plant protein drinks was a mere 13.1%. Consequently, consumers should use caution and carefully consider the nutritional details and ingredient information when selecting these drinks. Soybean (Glycine Max L. Merrill) is widely consumed as a dietary source of high-quality proteins, low quantity of saturated fat, dietary fiber, numerous nutritionally functional compounds, including phytosterols, isoflavones, saponins and anthocyanins. Soy milk can represent an animal milk substitute, while, soy protein isolate (SPI) is a high protein promising protein source (Kang et al., 2017; Rizzo & Baroni, 2018). These constituents offer diverse health-enhancing advantages, rendering the soybean distinct from other legumes. Despite the various health benefits, soybean consumption is limited due to poor content in some essential nutrients, unpleasant bean flavor, bitter taste, and interference with anti-nutritional factors. Therefore, soybeans-based foods were fortified, processed, cooked, or treated with heating, fermentation, and germination (Chung, Oh & Kim, 2017; Prabakaran et al., 2018).

A recent market survey of soy beverages showed that the predominant category was calcium and vitamin-fortified drinks which represented 60% of the market size. To narrow the nutritional gap with cow’s milk and meet essential dietary needs, plant-based drinks are often supplemented or fortified with calcium as soy milk has a relatively low calcium content (25 mg/100 mL) compared to cow’s milk (120 mg/100 mL) (Olías et al., 2023). The RDA for adults is between 1,000–1,200 mg daily, depending on age. For this reason, it is advised that vegans monitor their calcium status and consume calcium-fortified foods or calcium-rich food products or use calcium supplements (Volpe, 2007). Calcium absorption is governed by three main key aspects; the concentration of ionized calcium, the rate of absorption of ionized calcium and the transit time of the material through the intestine. Calcium chloride provides more elemental calcium per gram than other commonly used calcium salts and has high solubility which is independent of pH. Dietary calcium present in food matrices often contains calcium salts or complexes of calcium with proteins (Shkembi & Huppertz, 2022). Tocopherols (vitamin E) are natural antioxidants that increase the stability of fat-containing foods with functional health benefits. They are classified into four types of isomers (α, β, γ and δ) differing in their antioxidant activities and food system stability where (α > β > γ > δ) (Ghosh et al., 2022). Since soybean seeds contain a low concentration of α-tocopherol, hence, vitamin E activity in soybean is enhanced by increasing the α-tocopherol content (da Rocha et al., 2012). Considerable efforts have been undertaken to develop acceptable encapsulation technologies for the optimal administration of vitamin E because of its incompatibility with aqueous-based food matrices and vulnerability to degradation by environmental conditions. These challenges restrict the absorption and bioavailability of fat-soluble vitamins (Raikos et al., 2021). Furthermore, it is evidenced that vitamin E plays a role in increasing bone density, prevents bone calcium loss by neutralizing antioxidants and vitamin E-deficient diet will result in bone damage, probably due to impaired calcium absorption (Naina Mohamed et al., 2012).

Encapsulation of bioactive substances using nano-embedded delivery systems is a necessary and effective means of improving their stability and bioavailability. Nanoencapsulation is a technique well acknowledged for enhancing stability, bioavailability, prevention of contact with the biological environment, and improvement of intracellular absorption and activity of bioactive substances. This technology has enabled the reconfiguring a diverse range of food products and novel features that enhance their functional bioactivity. Soy protein isolate (SPI) is available, renewable, inexpensive, biodegradable, contains a variety of essential amino acids and beneficial physiological components, which can lower cholesterol levels and reduce the risk of cardiovascular diseases. In the last 3 years, the number of studies in each year reached more than 1,000, indicating that the attention to soybean protein nanoparticles has increased annually, which reflect its role as a hot topic in the research field (Katouzian & Jafari, 2016; Darwish et al., 2021; Gong et al., 2024).

Hence, this study aimed to compare the nutritional value of soybean milk, flour, and soy protein isolate (SPI), the preparation and characterization of vitamin E (α−tocopherol) impaired with/calcium salt (calcium chloride)/soybean protein isolate (SPI) nanoparticles (ECSPI-NPs). Then, application of the prepared nanoparticles to fortify soybean milk and compare their fortification effects on functional properties, color, and consumer acceptance with plain soybean milk (control) and the free form fortifications in order to develop soybean beverages with high nutritional value, maintaining the sensorial acceptance.

Materials and Methods

Chemicals and materials

Soybean seeds/flour packaged in Egypt by “Elarabia for packaging foodstuffs” was purchased from the local market in Alexandria Governorate, Egypt. Calcium chloride and α−Tocopherol (vitamin E) were purchased from Sigma-Aldrich Co. (St. Louis, MO, USA). The reagents and solvents used in this study were of analytical grade.

Soybean protein isolation

One hundred g of soybean flour was extracted with distilled water (10:1 water/soy flour) at 60 °C/30 min, pH 8.5 using NaOH (2N), then centrifuged (3,000 rpm/30 min) at 15 °C. The supernatant was acidified by HCl (2N) to precipitate the protein (20 °C/2 min, pH 4.5), refrigerated (4 °C/1 h), then centrifuged at (3,000 rpm/30 min) at 4 °C. The isolated crude protein was neutralized (pH 7) and freeze-dried (−20 °C/5 d). Figure 1 illustrates Soybean protein isolation steps according to Wang, Johnson & Wang (2004).

Figure 1 Soybean protein isolation.

Chemical composition

The chemical composition of soybean flour/soybean milk products, total solids, protein, and ash were assessed as described in Official Methods of Analysis of the Association of Official Analytical Chemists (AOAC, 2016). The carbohydrate content was quantified using the phenol-sulfuric technique, as outlined by Dubois et al. (1956) which measures the total hydrolyzable carbohydrate. The concentrations of calcium (Ca), potassium (K), sulfur (S), iron (Fe), and phosphorus (P) were quantified using atomic absorption spectroscopy (AAS) according to Olsen & Sommers (1982), Beaty & Kerber (1993).

AnkomXT10 fat extractor (Ankom Technology, Macedon, NY, USA) was applied to determine crude fat content (ANKOM_XT15 Technology, 2010). Crude fat was calculated using the formula:

(1) %CrudeFat=100×W2−W3W1.

W1, original sample weight; W2, pre-extraction dried sample and filter bag weight; W3, dried sample and filter bag after extraction weight.

Crude fiber content was determined using Crude Fiber Analysis in Feeds-Filter Bag 128 Technique A2000-AOCS Approved Procedure Ba 6a-05 (Ankom Technology Corp., Macedon, NY, USA) (ANKOM_2000 Technology, 2010), and calculated according to the formula:

(2) %Crudefiber=100×W3−W1W2

where: W1, bag tare weight; W2, sample weight; W3, end weight of the sample + bag

Energy and daily values for nutrition labelling (DV%)

Energy densities for carbohydrate (4 kcal/g, 17 kJ/g), for protein (4 kcal/g, 17 kJ/g), and fat (9 kcal/g, 38 kJ/g) (Hall et al., 2012). Energy was calculated according to the equation:

(3) Energy(Kcal)=(Carbohydrate(g)∗4)+(Protein(g)∗4)+(Fat(g)∗9)

% daily values for nutrition labeling were calculated based on a daily intake of 2,000 calories, which has been established as the reference for adults and children 4 or more years of age according to FDA (2004).

Amino acid profile analysis for soybean flour/soy protein isolate

Identification of free amino acids by using amino acid analyzer was conducted as follows; 1 g of each defatted soybean flour/SPI was extracted by boiling under reflux with 50 mL of 50% ethanol 3 times (each time for 3 h). For clarification, the combined ethanolic solutions were filtered and treated with trichloroacetic acid solution (10%). The supernatant fluid was concentrated under reduced pressure to 5 mL. The residue was washed with distilled water. The volume of the filtrate was adjusted to 100 mL using distilled water. A total of 5 mL from the diluted sample were dried under vacuum at 70 °C, then dissolved in 5 mL loading buffer (0.2 N sodium citrate buffer pH 2). The sample was filtered through a 0.45 micropore filter and injected into the amino acid analyzer (Pellet & Young, 1980).

The hydrolyzed protein amino acids were determined according to the method described by Jajic et al. (2013) as follows: Defatted powder (0.1 g) was digested with 10 mL of 6 N HCl in a sealing tube. The mixture was hydrolyzed at 110 °C for 24 h, and then filtered, and the hydrolyzed protein-amino acids were obtained by evaporation of the hydrolysate till dryness. The residue was washed with distilled water. The volume of the filtrate was adjusted to 100 mL using distilled water. The investigation of protein-amino acids was completed as in free amino acids. System of amino acid analyzer: Beckman system 7300 high-performance analyzer: Column: Na high-performance column 25 cm. Injected volume: 50 µl L. Detector: visible light detector. Retention time and separated area were obtained using Hewlett Packard 3390 recording integrator.

Determination of fatty acid content using gas-liquid chromatography

The analysis of fatty acids methyl esters was conducted using gas-liquid chromatography utilizing the Hewlett Packard Model 6890 chromatograph, as outlined by Schumann & Siekmann (2000). The experiment was carried out under the specified circumstances. The separation process was conducted using an INNO wax (polyethylene glycol) model No. 19095 N-123, with a maximum operating temperature of 240 °C. The capillary column used had dimensions of 30.0 m × 530 μm × 1.0 μm and a nominal flow rate of 15 mL/min, resulting in an average velocity of 89 cm/sec. The pressure during the separation was maintained at 8.2 psi. Column temperature was set to 240 °C using a temperature programming method. The programming included starting at initial temperature of 100 °C and gradually increased by 10 °C each minute until reaching the maximum temperature of 240 °C. The temperature was then maintained at 240 °C for 10 min. The injection temperature was set at 280 °C, using a back-inlet configuration. The split ratio used was 8:1, with a split flow rate of 120 mL/min. The gas saver was implemented at a 20 mL/min flow rate. The carrier gas used was nitrogen, with a 15 mL/min flow rate. The flame ionization detector temperature was set at 280 °C. The flow rate of hydrogen was 30 mL/min. The airflow rate was 300 mL/min.

Phenolic content (HPLC)

The separation and quantitative measurement of polyphenol content in soybean flour were conducted using the methodology outlined by Croci, Cioroiu & Lazar (2009). The HPLC equipment used was the Agilent 1260 model, manufactured by Agilent Technologies in California, USA. The separation procedure was conducted using the SUPLCO Discovery® BIO Wide C18 column, with dimensions of 4.6 mm × 250 mm and a particle size of 5 μm. The mobile phase comprised of two components; buffer solution containing sodium phosphate dibasic and sodium borate, with pH value of 8.2 (referred to as component A), and a mixture of acetonitrile (ACN), methanol (MeOH), and water in a ratio of 45:45:10 (referred to as component B). The mobile phase was delivered at a flow rate of 1.5 mL per minute. The mobile phase was sequentially programmed using a linear gradient. The column temperature was kept constant at 40 °C.

Preparation of vitamin E/calcium/soy protein isolate nanocapsules

Hydration of SPI powder (10 mg/mL) in deionized water was carried out with constant stirring for 30 min at 23 °C. After 30 min of standing time,the suspensions were adjusted to PH 12 with 0.5 M NaOH and stood for another 30 min. The alkaline-treated SPI solutions were heated at 85 °C for 30 min in securely closed tubes using a water bath to promote SPI unfolding and destroy big particles. The pH was brought back to 7.5 with 0.5 M HCl after the solutions cooled to 23 °C for 30 min. Protein was diluted to a final concentration of 30 mg/mL. Calcium was encapsulated by complexation, and Ca-nanoparticles were prepared by adding CaCl2 to the protein solutions at 10, 20, and 30 mM concentrations/10 g SPI. Also, vitamin E was dissolved in absolute ethyl alcohol (0.01%, w/v) and stirred for 30 min. Then, vitamin E was added to encapsulate at 50, 100, and 150 mg/g SPI concentrations drop by drop to 100 mL of the SPI solution at 25 °C. A PH of 7.5 was achieved in the final suspension. Using a VCX800 (Vibra Cell, Sonics, Newtown, CT, USA) and a 13 mm diameter probe (high-grade titanium alloy), we subjected the mixes to high-intensity ultrasound at an amplitude of 40% for 5 min in an ice bath to achieve homogeneity nanocapsule (Gaber & Barakat, 2019).

Characterization of ECSPI-NPs

Encapsulation efficiency

To extract unembedded ingredients from (supernatant); 10 mL of each sample was centrifuged at 8,000 g/30 min at 4 °C. To acquire unembedded vitamin E for vitamin E samples, the supernatant was mixed (1:1, v/v) with n-hexane. The absorbance value was then determined using a UV spectrophotometer at 285 nm. Additionally, newly generated samples were diluted in PBS before being combined with n-hexane and utilized as test samples. The samples were used to measure in a nitrous oxide-acetylene flame. The supernatant was taken to assess the calcium encapsulation effectiveness using atomic absorption spectrometry (Perkin Elmer model 703, Waltham, MA, US) (Gaber & Barakat, 2019). Encapsulation efficiency (EE%) was calculated according to the following equation:

(4) Encapsulationefficiency(EE%)=1−AmountofunembeddedTotalamountadded×100.

Measurement of particle size and ζ-potential

The nanoparticles’ ζ-potential and particle size and polydispersity index (PDI) were determined using a nanometer particle sizer Mastersizer 2000. Before measurement, the nanoencapsulation samples were diluted to 1 mg/mL protein concentration using 0.01 mol/L PH 7.5 phosphate buffers. The sample equilibration duration was 2 min, and the measurement temperature was 25 °C (Gaber & Barakat, 2019).

Transmission electron microscopy

The preparation of nanocapsules containing vitamin E, calcium, and soy protein isolate (ECSPI-NPs) was conducted for transmission electron microscopy (TEM) analysis. The samples underwent a dilution process using deionized water at a ratio of 1:100 (volume to volume). A small quantity of the diluted suspension was applied onto the electron microscopy grid coated with a format. It was allowed to remain undisturbed for 1 min, after which a solution of phosphotungstic acid (with a concentration of 2% at a pH of 7.2) was introduced. The grid was subjected to air-drying and afterward analyzed using a JEOL JEM-1400 plus TEM operating at a voltage increase of 100 kilovolts at an imaging level of 200,000 X (Soliman et al., 2019).

Cytotoxicity on vero cell line via viability assay-MTT protocol

A full monolayer sheet formed on the 96-well tissue culture plate after being infected with 1 × 105 cells/mL (100 ul/well) and incubated at 37 °C for 24 h. The growth medium was removed from the 96-well microtiter plates after forming a confluent sheet of cells. The cell monolayer was then washed twice with wash media. The examined sample was subjected to two-fold dilutions using RPMI medium supplemented with 2% serum, which served as the maintenance medium. A volume of 0.1 mL from each dilution was used for testing in separate wells, whereas three wells were designated as control and received just maintenance medium. The plate was incubated at a temperature of 37 °C and then subjected to examination. The cells were examined for any observable toxicity indications, such as partial or complete loss of the monolayer, cellular rounding, shrinkage, or the formation of cell granules. A solution of MTT was produced at a concentration of 5 mg/mL in phosphate-buffered saline (PBS) using a product provided by Bio Basic Canada Inc. (Ontario, Canada). Twenty microliters of MTT solution were introduced into each well and then placed on a shaking table at 150 revolutions per minute for 5 min. This procedure was conducted to ensure proper mixing of the MTT solution with the medium. Subsequently, the cells were incubated at 37 °C in a 5% CO2 atmosphere for 4 h, facilitating the metabolic processing of MTT. Following this incubation period, the medium was discarded, and the plate was carefully dried using paper towels to eliminate any residue if deemed required. The formazan, a metabolic product of MTT, was resuspended in 200 μL of dimethyl sulfoxide (DMSO). The suspension was then agitated on a shaking table at a speed of 150 revolutions per minute (rpm) for 5 min to ensure complete mixing of the formazan with the solvent. The optical density measurement was conducted at a wavelength of 560 nm, and the obtained values were adjusted by subtracting the background signal recorded at 620 nm. There should be a direct correlation between optical density and cell quantity (Mosmann, 1983). Vero cell lines were obtained from Science Way (Cairo, Egypt) for scientific researches and consultations, who also conducted the cytotoxicity viability assay-MTT protocol.

Soybean milk products’ preparation

After several preliminary experiments, soybean milk products enriched with calcium salts and vitamin E either in free form or in nano-encapsulated form were produced as described in Fig. 2, according to Yu et al. (2021). Soybean seeds were soaked in distilled water (1:9 w/w) at 20 °C/16 h. The soaked soybean was then coarsely grounded with food processor (Mienta chopper plus combi, France).

Figure 2 Soybean milk formulations.

The resulting slurry filtered using 100 mesh sieves to separate the soymilk from the residue. Then, the raw soymilk was preheated in a boiling water bath for 10 min and transferred to the electric stove for boiling at 100 °C/5 min. This is the traditional method used by the Chinese population for preparing soymilk. Table 1 exhibited the six enriched soybean milk formulations in addition to the control plain soybean milk for comparison.

Table 1 Soybean milk formulations.

Formulations	Acronym	Description	
Formula 1	CSM	Plain soybean milk (control)	
Formula 2	NEM	Soybean milk fortified with (ESPI-NPs)	
Formula 3	NCM	Soybean milk fortified with (CSPI-NPs) (20 mM/10 g SPI)	
Formula 4	NECM	Soybean milk fortified with (ECSPI-NPs) (100 mg/g SPI)	
Formula 5	FEM	Soybean milk fortified with free form of vitamin E (10 mg/100 mL)	
Formula 6	FCM	Soybean milk fortified with free form of calcium salt (80 mg (0.221 g CaCl2)/100 mL)	
Formula 7	FECM	Soybean milk fortified with free form of vitamin E + calcium salt (1:1)	
Note:

ESPI-NPs, SPI with vitamin E nanoparticles; CSPI-NPs, SPI with calcium salt nanoparticles; ECSPI-NPs, SPI with vitamin E and calcium salt nanoparticles.

Flavonoid, phenolic content and antioxidant activity (DPPH)

Total flavonoid content was assessed via colorimetric method as described by Sakanaka, Tachibana & Okada (2005). Distilled water (1.25 ml) was added to an aliquot of 250 μl of sample with 75 μl of 5% sodium nitrite followed by the addition of 150 μl of 10% aluminum chloride after 5 min. Six min later; 0.5 ml of 1 M sodium hydroxide was added then distilled water to reach volume of 3 mL. T80 UV/VIS spectrophotometer (PG Instrument Ltd., Loughborough, Leicestershire, UK) was used at 510 nm to read absorbance. A standard curve was plotted using different concentrations of catechol/ methanol (20–200 μg/mL). The results were expressed as mg of catechol equivalent per g of sample.

The determination of the total phenolic content was conducted using the Folin–Ciocalteu technique (Singleton, Orthofer & Lamuela-Raventos, 1999). Aliquate of 0.1 mL of Folin-Ciocalteu reagent was added to 2 mL of reconstituted extract. The mixture was allowed to stand for 15 min. Then, 3 mL of saturated sodium carbonate 2% (Na2CO3) was added. The mixture was allowed to stand for 30 min at room temperature and the total phenolic content absorbance was determined at 760 nm using T80 UV/VIS spectrophotometer (PG Instrument Ltd., Loughborough, Leicestershire, UK). Gallic acid was used as a standard. Total phenol values are expressed as mg of gallic acid equivalent (GAE)/g sample using the linear regression equation obtained from the standard gallic acid calibration curve.

The 2,2-diphenyl-1-picrylhydrazyl (DPPH) assay, the anti-radical activity was assessed by reaction with the radical DPPH dissolved in absolute ethanol. The reduction of DPPH causes a decrease in absorbance at 517 nm measured by (T80 UV/VIS spectrophotometer, PG Instrument Ltd., Loughborough, Leicestershire, UK). Five hundred µl at various concentrations of samples with 375 µl ethanol and 125 µl of a DPPH solution (0.02% prepared in ethanol) were prepared. A blank was prepared by mixing 500 µl of samples with 500 µl ethanol. A control containing 875 µl ethanol and 125 µl DPPH was prepared. After incubation for 60 min in the dark, the absorbance at 517 nm is measured using T80 UV/VIS spectrophotometer, PG Instrument Ltd., Loughborough, Leicestershire, UK. The anti-radical activity was determined using the formula:

(5) DPPHradicalinhibition(%)=Acontrol+Ablank−AsampleAcontrol×100

IC50 (mg/mL) was calculated based on inhibitory concentration at which 50% of DPPH radical is scavenged (Brand-Williams, Cuvelier & Berset, 1995).

Color analysis

The color of soybean milk products was analyzed using a Hunter colorimeter Ultra Scan VIS. Hunter L, a, and b values were utilized to convey values. The L* value represents the lightness on a scale of 0 to 100, with lower values indicating black and higher values indicating white. The a* value represents the degree of red and green color, with higher positive values indicating a greater presence of red. Lastly, the b* value represents the degree of yellow and blue colors, with higher values indicating a greater presence of yellow (Hunter & Harold, 1976).

Sensory acceptability

A sensory evaluation of fresh soybean milk products was performed by 50 trained/untrained panelists, including 30 men and 20 women between the ages of 27 and 60 at the Department of Nutrition, High Institute of Public Health, Alexandria University, Egypt. The samples included CSM, Control Plain Soy Milk, NEM, nano-capsulated vitamin E fortified soy milk, NCM, nano-capsulated calcium salt fortified soy milk, NECM, nano-capsulated vitamin E + calcium salt fortified soy milk, FEM, free form vitamin E fortified soy milk, FCM, Free (Darwish, Khalifa & El Sohaimy, 2018). A 9-point Hedonic scale was used to judge the samples (ISO/IDF, 2009). This scale included the test parameters of flavor, body and texture, appearance and color, odor, and overall acceptability, along with a scale of ten categories as follows: one for extremely dislike, two for much dislike, three for moderate dissatisfaction four for slight dislike, five for neither dislike nor like, six for slight like, seven for moderate like, eight for much like, and nine for extremely like.

Statistical analysis

Data were analyzed using IBM SPSS software package version 23.0 (IBM Corp., Armonk, NY, USA). Quantitative data was described using mean ± standard deviation. Significance of the obtained results judged at the 5% level. T-test (ANOVA) used for normally distributed quantitative variables, to compare between more than two groups, and post hoc test (LSD) for pairwise comparison (IBM Corp, 2015).

Results and discussion

Chemical composition and nutritional evaluation of soybean

Table 2 illustrates the chemical composition and nutritional evaluation of soybean flour and milk. The calculated energy values for flour and milk were 313.02, 26.39 Kcal/100 g respectively. Results indicated that protein, carbohydrates and fat content of soybean flour were 40.50, 20.88, 7.50 g/100 g respectively, that fulfil 81.00, 6.96, 11.54 % of the daily requirement (DV). While protein, carbohydrates and fat content of soybean milk were 0.83, 0.03, 2.54 g/100 g respectively, that fulfil only 1.68, 0.01, 3.91% of the daily requirement (DV). This significant difference is due to the moisture content of soybean milk (88.50%) against (7.10%) in soybean flour. Obtained results are in agreement with previous findings (Kundu, Dhankhar & Sharma, 2018; Paul et al., 2020).

Table 2 Chemical composition and nutritional evaluation of soybean flour and milk.

Component	DV (g)	Soybean flour	Soybean milk	
Content (g/100 g)	DV%	Content (g/100 mL)	DV%	
Energy (Kcal)	--	313.02a	--	26.39b	--	
Nutrients						
Moisture	--	9.00b ± 0.63	--	88.50a ± 0.89	--	
Protein	50	40.50a ± 2.80	81.00a	0.83b ± 0.002	1.68b	
Carbohydrate	300	20.88a ± 0.041	6.96a	0.03b ± 0.001	0.01b	
Fat	65	7.50a ± 1.01	11.54a	2.54b ± 0.03	3.91b	
Fiber	25	3.75 ± 0.35	15.00	ND	ND	
Ash	--	7.10a ± 0.010	--	0.60b ± 0.002	--	
Notes:

Data was expressed as means ± SD.

Daily values (per 100 g) for nutrition labeling, calculated based on a caloric intake of 2,000 calories for adults and children 4 years or more.

Means with different superscripts in a raw are significantly different at p ≤ 0.05 level.

ND, Not detected.

On comparing the mineral content of soybean flour, milk, and SPI in Table 3, the results showed that the content was significantly different as the highest content of all tested minerals (S, Ca, K, Fe, P), was reported in soybean flour except for sulfur (1,675 mg/kg) which was significantly higher in SPI (1,750 mg/kg) due to the concentration of the sulfur-containing amino acids (Table 4). The main mineral in soybean based on concentration (Table 3), was K, P, Ca, S, and Fe (12,500, 3,750, 2,500, 1,675, 100 mg/kg), respectively in soybean flour. On the other hand, on comparing based on providing the daily requirements % DV, the order would differ to be Fe, P, K, Ca, and S, as 100 g of soybean flour can provide 55.56, 37.50, 35.71, 25, and 19.71 of % DV, respectively. These findings are consistent with those of Vanga & Raghavan (2018). Values may be varied due to different species or sources, as reported by Reyes-Jurado et al. (2023).

Table 3 Mineral content of soybean flour, milk, and SPI.

Minerals	DV (mg)	Soybean flour		Soybean milk		SPI		
(mg/Kg)	DV%	(mg/L)	DV%	(mg/kg)	DV%	
Sulfur, S	850	1,675b ± 5.50	19.71b	98c ± 2.29	1.15c	1,750a ± 9.81	20.59a	
Calcium, Ca	1,000	2,500a ± 9.98	25.00a	60c ± 2.80	0.60c	1,500b ± 10.0	15.00b	
Potassium, K	3,500	12,500a ± 12.2	35.71a	630c ± 10.05	1.80c	3,750b ± 15.1	10.71b	
Iron, Fe	18	100a ± 4.90	55.56a	4.6c ± 0.09	2.56c	65b ± 1.96	36.11b	
Phosphorus, P	1,000	3,750a ± 30.0	37.50a	220c ± 9.97	2.20c	2,000b ± 47.9	20.00b	
Notes:

Data was expressed as means ± SD.

DV%, Daily Values (per 100 g) for nutrition labeling, calculated based on a caloric intake of 2,000 calories for adults and children 4 years or more.

Means with different superscripts in a raw significantly differ at p ≤ 0.05.

Table 4 Amino acids profile of soybean flour, milk and SPI.

Amino acids	Symbol	1Pattern	Soybean flour	Soybean milk	SPI	
mg/g	2AAS	mg/mL	2AAS	mg/g	2AAS	
Essential amino acids (EAA)	
Histidine	His	12	10.66	88.84	4.74	39.50	15.77	131.44	
Threonine	Thr	7	14.37	205.25	6.49	92.70	18.64	266.22	
Valine	Val	10	15.56	155.58	6.82	68.19	21.78	217.82	
Methionine	Met*	N/A	4.34*	N/A	2.25*	N/A	5.81*	N/A	
Methionine (+cysteine)	Met + Cys	13	9.55	73.43	4.80	36.90	13.27	102.11	
Phenylalanine	Phe	N/A	17.75	N/A	8.51	N/A	29.39	N/A	
Phenylalanine (+tyrosine)	Phe + Tyr	14	29.78	212.75	14.15	101.10	47.61	340.04	
Isoleucine	Ile	10	14.49	144.92	6.45	64.48	22.12	221.20	
Leucine	Leu	14	27.56	196.84	12.71	90.81	42.45	303.21	
Lysine	Lys	12	20.52	171.03	9.29	77.42	31.19	259.90	
Non-essential amino acids (NEAA)	
Asp	Asp	N/A	40.41	N/A	19.87	N/A	68.69	N/A	
Glu	Glu	N/A	74.69	N/A	36.65	N/A	137.02	N/A	
Serine	Ser	N/A	18.60	N/A	9.20	N/A	29.69	N/A	
Glycine	Gly	N/A	14.50	N/A	6.71	N/A	21.64	N/A	
Arginine	Arg	N/A	25.79	N/A	12.51	N/A	44.93	N/A	
Alanine	Ala	N/A	15.45	N/A	7.08	N/A	19.99	N/A	
Proline	Pro	N/A	29.23	N/A	16.84	N/A	66.94	N/A	
Cystine	Cys	N/A	5.20	N/A	2.55	N/A	7.47	N/A	
Tyrosine	Tyr	N/A	12.04	N/A	5.64	N/A	18.21	N/A	
EAA			125.25		57.26		187.15		
NEAA			235.91		117.05		414.58		
Total AA			361.16		174.31		601.73		
Notes:

1 Adults’ pattern (mg/g protein) as described (FAO/WHO/UNU, 1985). Based on the maximum estimate of the amount of amino acids individuals need to consume to maintain nitrogen balance (FAO/WHO, 1973) and assuming a safe daily protein consumption of 0.55 g per kg (the average figure for men and women), (National Research Council. 1989. Recommended Dietary Allowances: 10th Edition. Washington, DC: The National Academies Press. https://doi.org/10.17226/1349).

2 AAS, Amino Acid Score (%); HAA, Hydrophobic amino acids.

* First, limiting amino acids.

Amino acid profile of soybean milk, flour, and protein isolate

Amino acids profile and amino acid scores AAS% of soybean milk, flour, and protein isolate SPI are represented in Table 4. The term “complete protein” refers to foods that contain all essential amino acids in the correct proportion to build protein in the body (Nehete et al., 2013). In particular for vegetarian/vegan diets, the amino acid composition of soybeans demonstrates the excellent quality of its protein as a rich source of complete protein with a balanced quantity of essential amino acids that achieves total protein sufficiency in adults (Darwish et al., 2023).

The main essential amino acids in soybean milk based on content were leucine, lysine, phenylalanine (12.70, 9.29, 8.51 mg/g), while methionine was the limiting amino acid with content of 2.25 (mg/g). On comparing based on achieving AAS% they would be; threonine, leucine, lysine that achieve (92.70, 90.81, 77.42%) of AAS% requirement values. Although the nine standard essential amino acids for humans are reported to be present in soybean including tryptophan (Singer et al., 2019), but due to the indole ring of tryptophan is degraded under the acid condition used for protein hydrolysis, tryptophan cannot be analyzed by standard amino acid analysis methods (Friedman, 2018). Considering the non-essential amino acids; glutamic, aspartic, and proline recorded a high content in soybean milk with values of (36.65, 19.87, 16.84 mg/mL). Comparing to soybean milk, higher contents of amino acids were observed in soybean flour due to less moisture content and SPI as concentrated soybean protein. The results are similar to those recorded by Kudełka, Kowalska & Popis (2021).

Fatty acids content of soybean

Table 5 shows the results of a gas-liquid chromatographic examination of the fatty acid composition of soybean flour and milk. The findings indicated four saturated fatty acids (SFAs) in the soybean flour’s lipids component: palmitic acid, stearic acid, docosanoic acid, margaric acid, arachidic acid arranged in descending order according to concentrations 14.33, 4.24, 2.05, 1.13, 0.31 g/100 g, respectively. While, margaric acid was absent in soybean milk to include four SFAs which are; palmitic acid, stearic acid, docosanoic acid, arachidic acid with concentrations of 13.2, 5.33, 1.31, 0.82 g/100 g, respectively. These findings were in agreement with Lima et al. (2004). According to prior research oleic acids, which had concentrations of 21.98% and 29.74% and 21.98% and 29.74% of total fatty acids of soybean flour and milk, respectively, which represent the monounsaturated fatty acids that exhibit an excellent taste (Besbes et al., 2004). Linoleic acid (Cl8:2n-6) content of soybean flour and milk was (56.7, 48.84%) respectively. UFAs: SFAs ratio of soybean recorded (3.57, 3.80) in both flour and milk respectively. Noteworthy that during the processing of soybean protein, polyunsaturated fatty acids can be catalyzed to form hydroperoxides which then degraded to produce aldehydes, ketones, alcohols, and volatile compounds responsible for the formation of the beany flavor (Yang et al., 2023). Obtained results matched what was stated by Prabakaran et al. (2018). SFAs should be differentiated on food labels since they are metabolically and physiologically unique to be considered when establishing recommendations for diet as a significant factor in determining health, as well as the ideal dosage or ratio of UFAs (Simopoulos, 2003, 2004). SFAs are blamed for causing coronary heart disease and cancer. The British Department of Health recommends a ratio of PUFA/SFA of greater than 0.45, while WHO/FAO experts have issued recommendations for a “balanced diet” in which the recommended ratio of PUFA/SFA is above 0.40 (Altman & Bland, 1994; Wood et al., 2008).

Table 5 Fatty acids content.

Fatty acid	Symbol	Soybean flour	Soybean milk	
Saturated fatty acids (SFAs)	
Palmitic acid	(Cl6:0)	14.33	13.2	
Margaric acid	(C17:0)	1.13	ND	
Stearic acid	(C18.0)	4.24	5.33	
Arachidic acid	(C20.0)	0.31	0.82	
Docosanoic acid	(C22.0)	2.05	1.31	
Unsaturated fatty acids (USFAs)	
Oleic acid	(Cl8:l n-9)	21.98	29.74	
Linoleic acid	(Cl8:2 n-6)	56.7	48.84	
SFAs		22.06	20.66	
UFAs		78.68	78.58	
MUFAs		21.98	29.74	
PUFAs		56.70	48.84	
PUFAs: MUFAs ratio		2.58	1.64	
UFAs: SFAs ratio		3.57	3.80	
Note:

SFAs, Saturated fatty acids; USFAs, Unsaturated fatty acids; MUFAs, Monounsaturated fatty acids; PUFAs, Polyunsaturated fatty acid.

Determination of phenolic compounds (HPLC)

The results (Table 6) showed that soybean flour, soy protein isolate, and soybean milk contain 13 polyphenol compounds, including coumaric acid, ellagic acid, vanillin, ellagic acid, naringenin, syringic acid, daidzein, rutin, and apigenin. In comparison, hesperetin and gallic acid are absent in soybean flour or milk. By comparing compounds’ retention periods to those of genuine standards examined under the same circumstances, those compounds were partly identified. Most of phenolic compounds were concentrated in SPI as they showed higher values; on the other hand, the least concentrations were recorded in soybean milk due to high moisture content. Ferulic and syringic acids were the main phenolic compounds in flour (508.74, 442.94 µg/g), and SPI (491.78, 346.64 µg/g), while chlorogenic and ferulic acid were the main in soybean milk (16.93, 13.28 µg/mL). The results are in agreement with Rodríguez-Roque et al. (2013). Comparing chemical composition of soybean flour and milk showed the differences as soy protein sources where; soy flour was rich in protein, dietary fiber, and minerals, which was decreased due to extraction into soymilk, while amino acid, fatty acids and phenolic compounds profiles were also affected, which was supported by Olías et al. (2023).

Table 6 Phenolic compounds profile of soybean (HPLC).

Phenolic compounds	Soybean flour (µg/g)	Soybean milk
(µg/mL )	SPI
(µg/g)	
Gallic acid	73.37	ND	101.98	
Chlorogenic acid	83.89	16.93	194.63	
Catechin	ND	ND	ND	
Methyl gallate	35.19	2.01	27.73	
Caffeic acid	ND	ND	ND	
Syringic acid	442.94	6.75	346.64	
Pyro catechol	215.35	0.00	276.75	
Rutin	15.49	1.18	27.09	
Ellagic acid	16.91	0.76	60.99	
Coumaric acid	7.77	0.37	18.74	
Vanillin	ND	ND	ND	
Ferulic acid	508.74	13.28	491.78	
Naringenin	22.07	0.10	30.66	
Daidzein	19.47	0.16	209.02	
Querectin	ND	ND	ND	
Cinnamic acid	2.66	0.06	3.83	
Apigenin	35.17	0.23	215.69	
Kaempferol	ND	ND	ND	
Hesperetin	ND	0.18	54.94	

Characteristics of nanocapsules

Table 7 displays the encapsulation efficiency of the particle size measurements and the Polydispersity index (PDI) of potential nano-dispersions. All the potential nanodispersions showed particles with diameters between (51.95 ± 3 and 433.20 ± 21 nm) according to the production procedures. All nanoparticles had PDI values between 0.349 and 0.523. This indicates homogeneity of the produced nanoparticles (Dragicevic-Curic et al., 2010). The size distribution of nanoparticles is a crucial component influencing their biocompatibility and bio-distribution in vivo (Lodhia et al., 2010). The size of SPI nanoparticles was significantly affected by the calcium content. Particle diameters increased with increasing calcium content at a stable pH. Increases in CaCl2 concentration from 10 to 30 mM resulted in a corresponding increase in particle size from 141.60 ± 13 to 433.20 ± 21 nm at a pH of 7.5. Calcium acts as a salt bridge that allows polypeptide chains to approach one another, and acts as a shield for negative charges on polypeptide chains as a divalent cation. Calcium promotes the formation of β-sheet structures to create SPI aggregates stabilized by hydrogen bonding. Hydrophobic interactions might be used to connect these aggregates to create nano-networks. The residues of aromatic amino acids could be buried inside the SPI network, surrounded by a more “non-polar” milieu (Zhang et al., 2012). Also, the addition of vitamin E significantly affected the size distribution of SPI nanoparticles. The particle diameter increased with increasing vitamin E content of 50, 100, and 150 mg, resulting in 27.06 ± 2, 78.05 ± 5, and 170.50 ± 15 nm, respectively. After homogenization pressure treatment, the SPI-VE nanoparticles exhibited a smaller particle size than the SPI. This phenomenon could be attributed to the synergistic effect of SPI and VE, which results in forming more compact structure through hydrophobic interactions. The reduction in particle size of the SPI-VE interaction was observed with the escalation of high-pressure homogenization pressure. Under high-homogenization conditions, it is possible for the protein to undergo unfolding and subsequently form complex aggregates through disulfide bonds (Wang et al., 2017).

Table 7 Particle size, Polydispersity index (PDI), zeta potential, and encapsulation efficiency (EE %) of nanoparticles preparations.

Nanoparticles	Size (nm)	Calculated PDI	ζ potential (mv)	Encapsulation Efficiency (%)	
SPI-NPs	51.95f ± 3	0.349c	−14.60b ± 4	---	
50 mg ESPI-NPs	27.06g ± 2	0.389c	−23.40d ± 5	76.24b ± 1.79	
100 mg ESPI-NPs	78.05e ± 5	0.397c	−23.50d ± 6	75.25b ± 2.01	
150 mg ESPI-NPs	170.50b ± 15	0.479ab	−17.20c ± 5	68.75c ± 1.95	
10 Mm CSPI-NPs	141.60d ± 13	0.462b	−12.01b ± 3	91.45a ± 2.45	
20 Mm CSPI-NPs	159.46c ± 16	0.493a	−9.38ab ± 3	89.95ab ± 2.87	
30 Mm CSPI-NPs	433.20a ± 21	0.523a	−7.73a ± 3	52.35d ± 3.45	
Notes:

Results are means ± standard deviation for triplicates.

Different superscripts indicate differences in the means (p < 0.05).

PDI, Polydispersity index; EE, Encapsulation efficiency; SPI-NPs, Soy protein isolate nanoparticles; ESPI-NPs, Soy protein isolate with vitamin E nanocapsules; CSPI-NPs, Soy protein isolate with calcium salt nanoparticles.

The stability of nanoparticles is contingent upon the equilibrium between repulsive and attractive forces during their nearness, which can be utilized to anticipate the enduring stability of a nano-dispersion. Table 7 presents evidence that the zeta potential of SPI nanoparticles fell within the range of −7.73 to −23.50 mV. While the alteration was insignificant for the calcium specimens, the pattern was discernible. The reduction in calcium concentration increased the surface charge of the nanoparticles. The observed phenomenon may be ascribed to heightened intermolecular repulsions among protein chains at a pH of 7.5. The addition of calcium induced partial unfolding of the SPI protein, resulting in the exposure of non-polar amino acids and an elevation in surface hydrophobicity. The formation of nanoparticles could be attributed to the developing hydrophobic interactions within protein aggregates. This phenomenon results in the sequestration of non-polar amino acids within the interior of SPI networks, ultimately leading to a reduction in the surface hydrophobicity of SPI.

The complexes containing entirely vitamin E exhibited encapsulation efficiency values of 76.24%, 75.25%, and 68.75% for 50, 100, and 150 mg concentrations, respectively. The study found that the complex embedded with CaCl2 demonstrated different amounts of EE for Ca when added with 10, 20, and 30 mM, with values of 91.45%, 89.59%, and 52.35%, respectively. The phenomenon can be attributed to the ability of the protein and vitamin E or calcium complex to inhibit intermolecular interactions of proteins, thereby preventing the formation of aggregates. This improves the ability of proteins to serve as hydrophobic, physiologically active materials, as reported by Ye (2008), Hadian et al. (2016). The sample containing SPI with 50 mg of vitamin E exhibited a greater degree of encapsulation efficiency in comparison to the other two samples containing 100 and 150 mg of vitamin E. This trend also observed with calcium 10 mM higher than 20 and 30 mM, which could be attributed to developing more organized and condensed core-shell structure at a pH of 7.5. Sonic homogenization may alter the structure of SPI and enhance its interaction with VE or Ca (Karimi et al., 2016).

Morphological characterization of nanoparticles

The obtained zero-loss TEM micrographs (Fig. 3) provided evidence that the SPI nanoparticles were effectively synthesized in all non-transparent solutions following the introduction of calcium or vitamin E. Conversely, no nanoparticles were detected in native SPI solutions, as depicted in Fig. 3A. The produced nanoparticles exhibited a spherical morphology and displayed nearly a consistent distribution of sizes, as illustrated in Figs. 3B–3D. The nanoparticles that underwent treatment with calcium and vitamin E exhibited a particle of dark-rimmed particles of soy protein isolate surrounding the vitamin E, which appears light gray. The same result was observed by Geng et al. (2023). However, the calcium nanoparticles appeared as particles surrounded by a halo of calcium ions. Nonetheless, the pre-treated SPI did not display such a configuration. The aggregation of protein molecules is likely caused by calcium, which shields the electrostatic repulsion between charged protein molecules. Zhang et al. (2012) noticed that calcium may act as a bridge between the negatively charged carboxylic groups on adjacent protein molecules. The distinctive nature of this particle has the potential to facilitate the proficient encapsulation of bioactive agents for their targeted or regulated release by these nanoparticles.

Figure 3 TEM micrographs of nanoparticles preparations.

(A) SPI solution. (B) SPI nanoparticles (SPI-NPs). (C) SPI with vitamin E nanoparticles (ESPI-NPs). (D) SPI with calcium salt nanoparticles (CSPI-NPs).

Cytotoxicity assessment of nanoparticles preparations

Cytotoxicity assessment on vero cell line of the fortified milk with the three nanoparticles preparations compared with the control plain milk, are exhibited in (Fig. 4). The main aim of cytotoxicity evaluation was to estimate the safe concentration for soybean milk fortification with these nanoparticles. Obtained results indicated that combination of calcium salts and vitamin E nanoparticles caused to decrease IC50 approximately to the half (202.0 ug/mL) compared with the individual Ca or vitamin E nanoparticles (439.25, 400.63 ug/mL) respectively. The ingested nanoparticles effects are reported to be dose dependent and are relative to accumulation and toxicity to distant organs especially liver (Ferdous & Nemmar, 2020). The toxicity is also related to factors such as particle size, zeta potential, surface group, and aggregation state, however, reports and regulatory organizations limited the term and the hazard to materials with particle size <100 nm (Gomaa et al., 2024). Consequently, the applied fortification concentrations of the three nanocapsules preparations did not exceed the least IC50 value (202 ug/mL).

Figure 4 Light microscopy morphological examination and IC50 assessment on vero cell for different concentrations of soybean milk formulations with nanoparticles preparations.

(A) Cells treated with plain soybean milk (control). (B) Cells treated with soybean milk with (ESPI-NPs). (C) Cells treated with soybean milk with (CSPI-NPs). (D) Cells treated with soybean milk with (ECSPI-NPs) ((ESPI-NPs), (CSPINPs) (1:1)-ESPI-NPs, SPI with vitamin E nanoparticles; CSPI-NPs, SPI with calcium salt nanoparticles; ECSPI-NPs, SPI with vitamin E and calcium salt nanoparticles.

Antioxidant properties of soybean milk formulations

Figure 5 represent phenolic, flavonoid content and antioxidant potentials of soybean milk formulations to differentiate between the impact of nanoparticles and free forms fortifications. Nano-encapsulated forms of vitamin E, calcium salt and their combination; NEM, NCM, NECM (70.83, 134.38, 149.49 μg/mL) showed significantly higher content in phenolic colorimetric test comparing with the free form fortifications FEM, FCM, FECM (74.17, 83.13, 72.08 μg/ mL), while both were higher than control plain soybean milk (CMS) (70.52 μg/mL) (Fig. 5A). Different pattern was shown in flavonoid as the higher content was vitamin E fortifications in nano-encapsulated form NEM (44.70 μg/mL), combination of free form FECM 44.70 μg/mL), and free form FEM (39.43 μg/mL) (Fig. 5A). Soybeans total vitamin E content is evaluated with 156.0 μg/g (Chung, Oh & Kim, 2017), the increase in vitamin E content (tocopherol) showed a strong correlation with the increase in phenolic, flavonoids content (Kramer et al., 2014; Chatziharalambous et al., 2023).

Figure 5 Phenolic, flavonoids content and antioxidant potentials of soybean milk formulations.

(A) Total phenolic total flavonoids contents are expressed as μg/mL sample. (B) Antioxidant potential represented as IC50 (mg/mL) (inhibitory concentration at which 50% of DPPH radical is scavenged). Results represent means of duplicates ± SD. Different letters indicate differences in the means (p < 0.05). CSM, Plain soybean milk (control); NEM, Soybean milk fortified with (ESPI-NPs); NCM, Soybean milk fortified with (CSPI-NPs); NECM, Soybean milk fortified with (ECSPI-NPs); FEM, Soybean milk fortified with free form of vitamin E; FCM, Soybean milk fortified with free form of calcium salt; FECM, Soybean milk fortified with free form of vitamin E+ calcium salt; ESPI-NPs, SPI with vitamin E nanoparticles; CSPI-NPs, SPI with calcium salt nanoparticles; ECSPI-NPs, SPI with vitamin E and calcium salt nanoparticles.

These results were reflected on the antioxidant scavenging potentials represented as IC50 (mg/mL) the inhibitory concentration at which 50% of DPPH radicals are scavenged (Fig. 5B). As the results revealed that the significantly highest antioxidant potentials with the least IC50 were pronounced in vitamin E fortified formulations; NEM, FEM, NECM (139.55, 219.29, 239.84 mg/mL). Additionally, the increases in the antioxidant potentials in calcium fortified soybean milk formulations NECM, NCM (239.84, 426.39), could be attributed to the complex formation associated between Ca and soybean protein resulting structural changes that increase the antioxidant capacity (de Morais et al., 2020).

Color analysis of soybean milk formulations

The color analyses of soybean milk formulations are illustrated in Table 8. Free calcium fortifications; FCM, FECM (66.70, 66.63) showed to be lighter by (0.89, 0.82) in color than control CSM (65.81), which may be relied to the presence of calcium salt in free form. The nano-encapsulated fortification forms; NEM, NCM, NECM (64.02, 63.69, 64.83) succeeded to control significant differences by (1.79, 2.12, 0.98) with the least lightness variations comparing to control (65.81). Masking the colors applying the nano-preparations was previously supported (El-Kholy, Soliman & Darwish, 2019). All soybean milk formulations were located in the area between –a and b that indicates yellowish green. NECM tended to record the highest yellow color intensity with b value of 5.36 followed by NCM 5.20 while NEM had the least value 2.95. These results indicate a darker yellowish color that matched the sensory evaluation (Fig. 6). The relationship of particle size to color is well known and is mathematically quantified by scattering theories. Such properties are affected by reduced dimensionality of nanoparticles, as in polymer nanocomposites, distributions of nanoparticles are used to tune the index of refraction (Bhagyaraj & Oluwafemi, 2018). The obtained results came in accordance with Zhang et al. (2012), Raikos et al. (2021).

Table 8 Color analysis of soybean milk formulations.

Formulations	L	a	b	
CSM	65.81b ± 0.50	−1.34cd ± 0.11	3.79c ± 0.10	
NEM	64.02d ± 0.07	−1.75ab ± 0.20	2.95e ± 0.50	
NCM	63.69d ± 0.01	−1.67a ± 0.02	5.20a ± 0.20	
NECM	64.83c ± 0.10	−1.18d ± 0.06	5.36a ± 0.06	
FEM	62.86e ± 0.30	−1.84a ± 0.03	2.96d ± 0.03	
FCM	66.70a ± 0.20	−1.51bc ± 0.30	4.48b ± 0.04	
FECM	66.63a ± 0.40	−1.59abc ± 0.04	4.47b ± 0.40	
Notes:

Results represent means of duplicates ±SD.

Different superscripts indicate differences in the means (p < 0.05).

CSM, Plain soybean milk (control); NEM, Soybean milk fortified with (ESPI-NPs); NCM, Soybean milk fortified with (CSPI-NPs); NECM, Soybean milk fortified with (ECSPI-NPs); FEM, Soybean milk fortified with free form of vitamin E; FCM, Soybean milk fortified with free form of calcium salt; FECM, Soybean milk fortified with free form of vitamin E+ calcium salt; ESPI-NPs, SPI with vitamin E nanoparticles; CSPI-NPs, SPI with calcium salt nanoparticles; ECSPI-NPs, SPI with vitamin E and calcium salt nanoparticles.

Figure 6 Sensory evaluation of soybean milk formulations.

Data are expressed as means (n = 50) CSM, Plain soybean milk (control), NEM, Soybean milk fortified with (ESPI-NPs); NCM, Soybean milk fortified with (CSPI-NPs); NECM, Soybean milk fortified with (ECSPI-NPs); FEM, Soybean milk fortified with free form of vitamin E; FCM, Soybean milk fortified with free form of calcium salt; FECM, Soybean milk fortified with free form of vitamin E+ calcium salt. Asterisks (*) indicates differences in the means (p < 0.05). ESPI-NPs, SPI with vitamin E nanoparticles; CSPI-NPs, SPI with calcium salt nanoparticles; ECSPI-NPs, SPI with vitamin E and calcium salt nanoparticles.

Sensory evaluation of soybean milk formulations

The sensory evaluations of soybean milk formulations are illustrated in Fig. 6. The results revealed that the ECSPI-NPs with vitamin E and calcium salt nanoparticles significantly enhanced taste, texture and overall acceptability comparing to control CSM, but did not succeed to totally mask the soybean beany flavor as described by panelists that caused the low taste scores recorded in all fortifications. The main reason for the beany flavor in soybean protein is that the polyunsaturated fatty acids; linoleic acids (Table 5), are oxidized to form hydroperoxides under the action of lipoxygenase, which are then degraded to aldehydes, ketones, and alcohols (Yang et al., 2023). These results were in agreement with Kundu, Dhankhar & Sharma (2018) and Paul et al. (2020).

Conclusion

In conclusion, comparing the nutritional value of soybean milk, flour, and SPI indicated that the soybean is a good source of protein as that soybean flour protein content fulfilled 81% of DV%, with good quality as complete protein considering its essential amino acid content especially threonine, leucine, lysine that provided 92.70, 90.81, 77.42 of AAS% requirement values, respectively, taking into account that moisture content of soybean milk caused lower nutrients concentrations. Its healthy fat content represented in unsaturated fatty acids (USFs); omega-6 and omega-9 content that represented 21.98, 56.7% of total fat in flour. Ferulic acid was the main phenolic compound in soybean flour, milk and SPI. The prepared calcium and vitamin E nanoparticles (ECSPI-NPs) exhibited that they were effectively synthesized under TEM, stability and safety up to IC50 value (202 ug/mL) on vero cell line, which was considered in the application concentrations. ECSPI-NPs fortification enhanced phenolic content, color, texture and consumer overall acceptance. Findings supported the use of vitamin E and calcium salt nanocapsules in food applications taking into consideration their safety. Obtained results can be considered as future sustainable diets which are outlined by the FAO as those that “contribute to food and nutrition security and a healthy life for present and future generations while having low environmental implications” (FAO, 2010; FAO/WHO, 2019). Sustainable products increase natural and human resources while providing suitable amount of nutrients. Additionally, they are affordable, accessible to all, economically egalitarian, and protective of ecosystems and biodiversity. In the future, the effectiveness of soybean protein isolate nanoparticles as packaging carriers, stability, bioavailability and delivery functions to bioactive substances especially in enhancing plant-based beverage quality; needs further evaluation either in vitro or in vivo. This would help to establish greener and safer industrial chain of soybean protein nanoparticles for potential applications in food and medicine.

Supplemental Information

Supplemental Information 1 Raw data.

Additional Information and Declarations

Competing Interests

Author Contributions

Patent Disclosures

Data Availability

The authors declare that they have no competing interests.

Heba A. I. M. Taha conceived and designed the experiments, performed the experiments, analyzed the data, prepared figures and/or tables, authored or reviewed drafts of the article, and approved the final draft.

Neveen F. M. Agamy conceived and designed the experiments, analyzed the data, authored or reviewed drafts of the article, and approved the final draft.

Tarek N. Soliman conceived and designed the experiments, performed the experiments, analyzed the data, prepared figures and/or tables, authored or reviewed drafts of the article, and approved the final draft.

Nashwa M. Younes analyzed the data, authored or reviewed drafts of the article, and approved the final draft.

Hesham Ali El-Enshasy analyzed the data, authored or reviewed drafts of the article, funding, and approved the final draft.

Amira M. G. Darwish conceived and designed the experiments, performed the experiments, analyzed the data, prepared figures and/or tables, authored or reviewed drafts of the article, and approved the final draft.

The following patent dependencies were disclosed by the authors:

This work was registered in Patent Office, Academy of Scientific Research and Technology ASRT, Cairo, Egypt, No. 1002/2023/P/EG.

The following information was supplied regarding data availability:

The raw measurements are available in the Supplemental File.

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
