# Peer review of "Preparation and characterization of vitamin E/calcium/soy protein isolate nanoparticles for soybean milk beverage fortification"

_PeerJ, doi:10.7717/peerj.17007_

## Round 0.1 · original submission · Major Revisions

Please authors, kindly address the concerns of the reviewers. Make the best effort to provide the details in your response both in the revised manuscript, as well as the reviewers' comments.

**Language Note:** The review process has identified that the English language must be improved. PeerJ can provide language editing services - please contact us at copyediting@peerj.com for pricing (be sure to provide your manuscript number and title). Alternatively, you should make your own arrangements to improve the language quality and provide details in your response letter. – PeerJ Staff

·

Basic reporting

1. The background of this study is still unclear. Why should the researchers compare soymilk, flour, and protein? The reason for vitamin E fortification is also unclear. Researchers need to provide a more detailed background in the introduction of the study.
2. What is the novelty of this study? The authors need to explain in the introduction section
3. Line 104, "then centrifuge centrifuged" this sentence is quite ambiguous.
4. References used in the study are quite old and need to be updated.
5. The authors explained the data very well

Experimental design

1. In line 91, researchers stated that the calcium source used in the study was calcium chloride. Why calcium chloride was chosen in the study, while there are other calcium salts that have less impact on soymilk sensory characteristics?
2. Could authors specify the status of panelists in sensory evaluation? Were they trained or untrained panelists?

Validity of the findings

1. In section 3.1. Chemical composition and nutritional evaluation of soybean (Line 297 - 306), authors have explained very well. However, it would be great if authors could explain their reason for comparing soybean flour with soymilk because those two products are extremely different and incomparable.
2. The same things go in sections 3.2 to 3.4. Authors need to specify the reason for comparing soybean flour and soybean milk and not only explain the data.
3. In Table 8, It would be great if the authors could also calculate the color difference of fortified soymilk to control soymilk.
4. The data in Figure 6 are better explained in a table so readers can understand the trends and differences in the data.
5. Is there any significant results on the sensory evaluation data? (Line 498 - 503). The authors need to provide the statistical results.

Additional comments

The authors have successfully explained the data of the study. However, the reason for comparing soybean flour and soymilk is not well defined. Comparing the nutritional composition of soybean-based food is not suitable to show the benefits of soybean-based products.

The authors need to revise the introduction of the study so readers can easily understand the purpose of the study.

Reviewer 2 ·

Basic reporting

The manuscript is interesting, however, it contains several issues mainly the language, thus, a fluent English speaker must revise it.

Please address the following points:
1. Remove the abbreviations from the title ((ECSPI-NPs).
2. L32-35: the aim is non-clear, it msut be matched well with the title.
3. L 36: DV? is corerct??
4. L 39: wherre the full word of this abbreviation?? AAS
L42: specify the ratio of these fortification.
- The abstract should be improved and the focus should be on the main insights of the developed milk. Also, what about the sensory score of the fortification?
-Keywords: Phenolic content (HPLC)??
-L 263: 2.12. Flavonoid, phenolic content and antioxidant Activity should be wrote in details.

Experimental design

What about the characterization of the stability of the beverage either fresh or stored one?

Validity of the findings

-Write the limitations and future perspectives in the Conclusions section.
-The stability index should be provided.

---

## Round 0.2 · Major Revisions

Authors, thank you for your patience.

Concerns are raised, which have to be addressed:

What about the characterization of the stability of the beverage either fresh or stored? In addition, you have replied only to Reviewer 3. Kindly provide adequate responses to these because stability is the main target of the beverage quality.

In the introduction, kindly add more information about why calcium and soy protein are relevant to enhancing beverage quality. Also add, what is the essential role of fortification, and why is it useful?

Please, arrange the introduction to be a maximum of three paragraphs.

In the results and discussion, please make an effort to provide more in-depth explanations when discussing data, do not just state literature, tell us more of the why.

Conclusions, please add direction for future research.

Reviewer 2 ·

Basic reporting

Non be revised according to my comments.

Experimental design

Again: What about the characterization of the stability of the beverage either fresh or stored one?

Validity of the findings

The stability is the main target of the beverage quality.

---

## Round 0.3 · accepted · Accept

After a very thorough check of the revised manuscript, I am very satisfied with this current version. The peer review process enhanced the authors' capacity to elevate the quality of the work. Thank you, authors, for finding PeerJ as your journal of choice, and look forward to your future scholarly contributions.
Congratulations

Reviewer 2 ·

Basic reporting

The manuscript can be accepted.

Experimental design

Fine

Validity of the findings

Fine